# A Comparative Analysis of Tumors and Plasma Circulating Tumor DNA in 145 Advanced Cancer Patients Annotated by 3 Core Cellular Processes

**DOI:** 10.3390/cancers12030701

**Published:** 2020-03-16

**Authors:** Kristian Larson, Radhamani Kannaiyan, Ritu Pandey, Yuliang Chen, Hani M. Babiker, Daruka Mahadevan

**Affiliations:** 1University of Arizona College of Medicine, 1501 N Campbell Ave, Tucson, AZ 85724, USA; kris7c@email.arizona.edu; 2Vidant Medical Center, 2100 Stantonsburg Rd, Greenville, NC 27834, USA; radha.maran@gmail.com; 3Department of Cellular and Molecular Medicine, University of Arizona Cancer Center, 1515 N Campbell Ave, Tucson, AZ 85724, USA; ritu@email.arizona.edu; 4University of Arizona Cancer Center, 1515 N Campbell Ave, Tucson, AZ 85724, USA; yuliangchen@email.arizona.edu; 5Early Phase Clinical Trials Program, University of Arizona Cancer Center, 1515 N Campbell Ave, Tucson, AZ 85724, USA; hanibabiker@arizona.edu

**Keywords:** molecular targeted therapy, drug resistance, neoplasm, high-throughput nucleotide sequencing, DNA mutational analysis, liquid biopsy

## Abstract

Matched-targeted and immune checkpoint therapies have improved survival in cancer patients, but tumor heterogeneity contributes to drug resistance. Our study categorized gene mutations from next generation sequencing (NGS) into three core processes. This annotation helps decipher complex biologic interactions to guide therapy. We collected NGS data on 145 patients who have failed standard therapy (2016 to 2018). One hundred and forty two patients had data for tissue (Caris MI/X) and plasma cell-free circulating tumor DNA (Guardant360) platforms. The mutated genes were categorized into cell fate (CF), cell survival (CS), and genome maintenance (GM). Comparative analysis was performed for concordance and discordance, unclassified mutations, trends in *TP53* alterations, and PD-L1 expression. Two gene mutation maps were generated to compare each NGS platform. Mutated genes predominantly matched to CS with concordance between Guardant360 (64.4%) and Caris (51.5%). *TP53* alterations comprised a significant proportion of the mutation pool in Caris and Guardant360, 14.7% and 13.1%, respectively. Twenty-six potentially actionable gene alterations were detected from matching ctDNA to Caris unclassified alterations. The CS core cellular process was the most prevalent in our study population. Clinical trials are warranted to investigate biomarkers for the three core cellular processes in advanced cancer patients to define the next best therapies.

## 1. Introduction

Precision oncology strives to develop new targeted and immune therapies to improve overall survival (OS) [1]. Molecular profile-based clinical trials, including IMPACT [2] and WINTHER [3], have demonstrated a clear positive impact of matched-targeted therapies (MTT) against patient-specific gene alterations over chemotherapy. Small molecule inhibitors in various stages of development are designed to block key oncogenic signaling pathways. For example, BRAF and ALK inhibitors are examples that have demonstrated increased OS in melanoma and non-small cell lung cancer (NSCLC), respectively [4,5]. 

Studies have shown that tumor molecular profiles are multilayered and interactive. *TP53* mutations remain a clinical challenge and are associated with poor outcomes across many cancer subtypes [6,7,8]. PD-L1 status correlates to poor prognoses and predictive of responding to anti-PD-1 agents [9,10]. Breast adenocarcinoma (BAC) treated with PARP inhibitors up-regulating PD-L1 expression highlights the benefits of anti-PD-L1 therapy for this resistant state [11]. 

The development of MTTs that encompass complete molecular profiles is quintessential to personalized cancer treatments [2,12]. A review grouped a dozen regulatory signaling pathways into categories that reflect three fundamental cellular processes: cell fate (CF), cell survival (CS), and genome maintenance (GM) [13]. By categorizing the molecular profile into CF, CS, and GM, we aimed to integrate a comprehensive summary of driver and passenger mutations and display the corresponding tumor heterogeneity. We hypothesize that categorizing the mutational profile of each individual tumor to CF, CS, and GM will elucidate cellular processes (patterns) that provide a better understanding of tumor evolution and the development of drug resistance. In addition, since *TP53* is the most common mutated gene in a myriad of cancer subtypes, *TP53* is given special attention. Considering immune suppression is a key factor in modulating the tumor microenvironment, PD-L1 expression is also included in our analysis. By comparing next generation sequencing (NGS) platforms that assay tumor tissue and plasma circulating tumor DNA (ctDNA), we explored concordance versus discordance to discover tumor heterogeneity.

Patients’ genetic alterations are increasingly being revealed through a variety of NGS platforms. Interpretation and clinical decision-making of the results can be challenging. To address these issues, we present an integrated study of 145 patients enrolled in phase 1 clinical trials and are the first to compare 25 different cancer subtypes with data from two NGS platforms and gene category annotation. 

## 2. Results

### 2.1. Cell Survival (CS) Mutations Dominate Cell Fate (CF) and Genome Maintenance (GM) Mutations

NGS platforms detected a total of 173 mutated genes from 142 patients. These 173 mutated genes categorized to 53.2% (*n* = 92) CS, 37.6% (*n* = 65) CF, and 9.2% (*n* = 16) GM (Figure 1). The same trend CS > CF > GM followed at the platform level with CS 64.4% in Guardant360 and 51.5% in Caris (Table 1). 

When analyzed at the cancer subtype level, 15 of 25 cancer subtypes exhibited a trend of CS > GM > CF. Despite having fewer genes, GM contributed to more alterations than CF. Seven cancer subtypes also followed a trend of CS dominance, but CF and GM swapped positions. Only esophageal squamous cell carcinoma (ESCC) (*n* = 1) demonstrated a trend of GM dominance followed by CS and CF. Aberrations from these trends are observed in carcinoma of unknown primary (CUP) and neuroendocrine tumors (NET), which both represent limited patient sampling. Paired analysis using Fisher’s exact tests for these three cellular processes from results combined from both platforms showed no significant *p*-value indicating there is no association between these processes (Appendix A), and they occur independent of each other. Testing individual platform results show association between the occurrence of CS and CF (*p* = 0.008) on Caris platform and between CS and GM on both Caris (*p* = 6.9 × 10^−19^) and Guardant360 (*p* = 0.01). There was no significant association found between CF and GM on any platform. Patients were divided by their age (< 60-yr vs. > 60-yr) into two groups and these three processes were tested for prevalence in either of the age group (Appendix A). No association was found with age and occurrence of any of these three processes and no association was found with TP53 mutations. We tested this on both individual platforms and combined platform results. 

The trends demonstrated in the cancer subtypes generally agree between both platforms. In the cases of cholangiocarcinoma and prostate adenocarcinoma, there is platform discrepancy between the contributions of GM and CF. As described previously in the limited patient samples, platform trend disagreement was observed most significantly in ESCC, CUPS, and NETs but a larger dataset is needed for statistical confirmation.

### 2.2. TP53 is the Most Frequent Mutation

Guardant360 and Caris detected a total of 1005 and 524 specific alterations of all mutated genes, respectively. Of these, TP53 comprised a significant proportion at 13.1% (*n* = 132) and 14.7% (*n* = 77), respectively. Fifty-eight of these TP53 mutations matched at specific alteration level across the platform. Matched TP53 alterations in colorectal cancer (CRC) dominated 29.3% (*n* = 17), followed by pancreatic adenocarcinoma 17.2% (*n* = 10), and BAC 12.1% (*n* = 7). Platform-matched TP53 alterations appeared substantially in CRC; there were four of R175H and R273C, three of R248Q and R282W, and two of R196, R248W, and R273H alterations. The BAC also contained platform-matched TP53 alterations, including two of E285K and G245S. TP53 has the highest frequency of the mutations. We tested for two-way associations with the three cellular processes across both platforms for all patients (Appendix A). Our results showed that the GM process has a significant association with TP53 mutational status in patients (*p* = 2.2 × 10^−16^), however the CS and CF processes have no significant association with TP53 mutation status in patients. Patients divided into two groups by age (> 60-yr and < 60-yr) were tested for association with TP53 mutation status (Appendix A) with no association found. 

### 2.3. Trend in PD-L1 Status

Twenty patients tested positive for PD-L1 by Caris immunohistochemical (IHC) staining (Figure 2). Although head and neck squamous cell carcinoma (HNSCC) and pancreatic adenocarcinoma contain the largest number of positive IHC stains (5 and 4, respectively), HNSCC (55.5%), TCC (50%), GIST (50%), and NET (50%) represent the greatest proportion of positive PD-L1 stains per cancer subtype. Pancreatic adenocarcinoma (15.4%) and CRC (3.4%) entailed the least positive PD-L1 stains per cancer subtypes.

### 2.4. Marked Discordance Across the Platforms

Overall, the data show significant discordance in gene mutations across the platforms (Figure 3). At the individual patient level, the mean discordance per patient was 5.3 (range: 0–39). No discordance was detected in six patients. The mean concordance was 1.54 per patient (range: 0–9). 

At the pooled genes level, 223 genes were concordant and 760 genes were discordant. TP53 represented the highest frequency of concordant gene 29.1% (*n* = 65) followed by APC 13.5% (*n* = 30), KRAS 10.3% (*n* = 23), and PIK3CA 4.5% (*n* = 10). Interestingly, discordance followed a similar trend with TP53 10.3% (n = 78), EGFR 5.8% (*n* = 44), KRAS 5.0% (*n* = 38), and PIK3CA 4.3% (*n* = 33). 

Discordant genes were stratified into the three core cellular processes resulting in CS (61%), CF (20%), and GM (19%). This trend was roughly comparable to the stratified mutations of the overall cancer subtypes.

### 2.5. Identification of Potentially Actionable Mutations

The Caris-MI/X NGS platform analyzes tumor-only exon mutations in oncogenes and tumor suppressors. In contrast, the Gaurdant360 NGS platform analyzes cfDNA in tumors versus normal donor volunteer whole exome sequencing (WES) (ages: 20–40-yr), i.e., reference normal DNA. Plasma cfDNA from patients with mutations can detect up to 0.1% mutant allele frequencies (MAFs) from a background of cfDNA extracted from healthy donors and reported as acquired somatic mutations by digital sequencing algorithms [14]. An actionable mutation is defined as a genetic aberration in the DNA (e.g., activating mutation) when detected in a patient’s tumor, and would be expected or predicted to affect a response to a targeted treatment available in basket or umbrella clinical trials, FDA-approved treatments, or be available for off-label treatment [15]. Guardant360 detected genes were found in 19 of 142 patients that matched an exact alteration in the Caris unclassified mutation section (GaDCUS) (Table 2). We found one matching alteration in 16 patients and several in the remaining three patients. GaDCUS appeared frequently in the CRC (21.1%). Also, four mutated genes appeared across multiple patients. ARID1A appeared in the BAC, CRC, and NSCLC. CDKN2A appeared in sarcoma and pancreatic adenocarcinoma groups. ALK appeared in CRC and HNSCC. NF1 appeared in CRC and pancreatic adenocarcinoma. For example, the ALK (F1408L, G1473E) are novel mutations and whether they are sensitive to ALK tyrosine kinase inhibitors is not known but needs further evaluation. Similarly, the AR (P135L, A810T) are also mutants needing further investigation (Table 2). 

## 3. Discussion

Our study characterized passenger and driver mutations from NGS in tissue-based and plasma ctDNA samples into the three core cellular processes of tumorigenesis [13]. A review of the literature comparing advanced cancer patients’ molecular profiles concurrently for tissue based (Caris MI/X) and plasma (Guardant360) by NGS with annotation to the three core cellular processes has not been described before. We identified that CS genes dominated compared to GM and CF genes in our study population. GM and CF genes were prevalent equally. Similar trends were maintained at each platform level as well. Paired analysis using Fisher’s exact tests for the three cellular processes combined from both platforms showed no significant P-value, indicating no association and that the processes were independent of each other. Testing individual platforms showed association between CS and CF (*p* = 0.008) on Caris and between CS and GM on both Caris (*p* = 6.9 × 10^−19^) and Guardant360 (*p* = 0.01). Patients divided by age (<60-yr vs. >60-yr) showed no association with *TP53* mutations or any of the three cellular processes. 

It can be surmised that tumor types with unfavorable growth conditions, such as hypoxia and hypoglycemia, result in selective mutations of genes such as *KRAS*, *BRAF*, *PIK3CA*, and *TP53* [15]. These altered pathways lend cancer cells survival advantages by employing strategies such as angiogenesis and *GLUT1* upregulation [15,16]. Further studies utilizing this conceptual framework through large-scale prospective studies in targeted and immune checkpoint therapy trials are required to validate our analysis. 

Patients with ESCC, NET, and CUP revealed mixed results that did not follow the predominant trend. However, these groups had the least number of patients and yielded low statistical power. A study that elucidated the genomic landscape of ESCC in 133 patients found the most frequent somatic mutations included *TP53* (93%), *CCND1* (33%), *CDKN2A* (20%), *NFE2L2* (10%), and *RB1* (9%) [17]. These driver mutations of ESCC predominantly belong to the CS and less to the GM and CF processes, which positively compares to our analysis. Innovative clinical trial designs that integrate molecular profiles to the three core pathways to select appropriate MTTs [18] may help prevent or overcome drug resistance. In addition, clinical decision-making about treatment selection would shift from single gene mutations to more comprehensive molecular profile-based approaches. Assessing the three core cellular processes may potentially renovate precision oncology and improve patient survival.

High frequencies of *TP53* mutations play a transformative role in tumorigenesis across multiple cancer subtypes [19,20,21]. Most patients had *TP53* mutations with predominance within the CRC group (29.3%). *TP53* mutations consequently resulted in the highest rates of concordance and discordance between the NGS platforms. Tumor responses to antiangiogenic drugs, such as bevacizumab, have indicated a link to *TP53* mutations as a biomarker [22]. Integrating data on specific *TP53* alterations with transcriptomics may help guide therapy in addition to a more comprehensive molecular profile. 

PD-L1 expression adds another layer of complexity to NGS molecular profiling. Studies have demonstrated aggressive cancer growth with defective anti-tumor immune responses and resistance by immunoediting of PD-L1 [23,24]. Understanding a patient’s molecular profile, including copy number amplifications (CNAs), may help predict drug resistance and consequently help tailor a regimen(s) more efficacious and less toxic to normal tissue.

Comparison of alterations detected by tissue based (Caris) versus plasma ctDNA (Guardant360) platforms exhibited marked discordance. Our study included mutations detected at low, intermediate, and high frequencies. These results support other studies that show marked discordance between platform comparisons and the inclusion of low alteration frequencies [25,26]. We included all frequency ranges to form a complete genetic profile to demonstrate the degree of intra- and inter-patient tumor heterogeneity. Since plasma ctDNA provides a snapshot or summary of all metastatic sites of cancer within a patient, comparing the detected mutations of a focused tissue biopsy can miss other relevant mutations. A study that conducted a saturation analysis of 21 tumor types concluded that genes with low frequencies should be included in analyses to better comprehend the full implications of defective signaling pathways [27]. Plasma ctDNA analyses have shown therapeutic benefits and can help understand tumor evolution, including mechanisms of resistance such as acquired *ESR1* mutations that induce aromatase inhibitor resistance [28] in BAC and EGFR resistance to 1st and 2nd generation EGFR tyrosine kinase inhibitors in NSCLC. This reinforces the practice of following multiple plasma ctDNA samples throughout patient management, especially before progression [29] prior to imaging. 

Although we chose the most recent tissue and plasma samples, following patient ctDNA samples at multiple intervals may offer some advantages. For example, a retrospective study of nine metastatic BAC patients demonstrated that more optimal therapies could have been chosen by following changes in ctDNA [29]. A recent joint review by the American Society of Clinical Oncology (ASCO) and College of American Pathologists (CAP) provided contrary evidence in the clinical utility of plasma ctDNA in the early detection of cancer, monitoring treatment or post-treatment residual disease [30]. Several factors influence ctDNA, which include low tumor burden, number of metastatic sites and timing of sample collection during active treatment and/or surgical resection [31]. As supported by a study that evaluated cancer driver genes, our study does not account for all tumor heterogeneity [32]. Guidelines on specimen collection, especially with plasma ctDNA, must be developed to yield consistent results among NGS platforms and to accurately characterize the genetic heterogeneity of cancer. Tissue-based biopsies have shaped approaches to MTT with improved patient outcomes; including ctDNA will likely confer the similar benefits in early phase clinical trials as demonstrated by the TARGET study [33,34]. 

Our study revealed potentially actionable alterations. Cross-comparison of NGS in tissue and ctDNA yielded 26 somatic mutations that previously were categorized as variants of unknown significance (VUS). Caris compares a patient’s sample to a database of known driver mutations to confirm pathogenic alterations [15]. Guardant360 captures the full spectrum of plasma cell-free DNA (cfDNA) and genetically distinguishes tumor vs. normal DNA to infer clinically relevant alterations [14]. Additionally, Caris assigns alterations that have an unknown growth advantage to the “unclassified” section. Hence, alterations detected by Guardant360 in the Caris unclassified section (GaDCUS) strongly suggest mutations that are somatic and potentially targetable. A study [35] that identified putative germline mutations in ctDNA reported detection of *APC, ATM, BRCA1/2, CDKN2A, MLH1, NF1, RB1, RET, SMAD4*, and *TP53*. We found these mutations in both our NGS platform analyses except *MLH1* in Guardant360. Our GaDCUS mutations matched *CDKN2A, NF1*, and *RB1* as well, which help discern germline and somatic mutations. Our patients’ GaDCUS mutations fell into the CS and CF categories approximately equally. We detected one GaDCUS mutation, *TERT (A670V)* in a CRC patient (#23), that resides in the GM category. Since *TERT* plays a major role in tumor cell immortality through telomere lengthening, this GaDCUS mutation may have revealed a potential driver that contributed to the pathogenesis of this CRC case [36]. A study of various tumor types identified over 50 gene candidates that mapped to interactive pathways of known major cancer driver genes [37]. By performing NGS on platforms that differ in methodology, we can identify clinically relevant alterations. Studies have utilized software tools such as CHASM and ANNOVAR to statistically determine the significance of driver and passenger gene mutations [38]. When mutations are discovered, these databases can compute more comprehensive analyses of cancer genomes and heterogeneity [39]. Additionally, more complex stratifications can be applied to determine the primary drivers of a patient’s tumor growth and guide selection of targeted and immune checkpoint therapies. 

### Limitations

Comparing NGS data of tumor biopsy (Caris) to plasma ctDNA (Guardant360) render both a comparative limitation and an illustration of the heterogeneity that exists in advanced cancer patients. This heterogeneity contributes to the significant discordance observed in our analysis. However, it demonstrates the variability of actionable mutations at tumor sites and unpredictable responses to MTT. Our study analyzed a snapshot of time as opposed to following the mutational evolution with time. Following plasma ctDNA samples in real time will help anticipate tumor evolution and provide an opportunity to switch therapy prior to imaging. Tissue-based samples entail greater costs and toxicity of procedure for the patient. Cancer subtype-specific characteristics, such as treatment history, were not accounted for in our diverse study population. Further delineation of the annotated trends should include these measures especially for the potential design of clinical trials.

## 4. Materials and Methods

### 4.1. Patient Selection, NGS Platforms, and Sample Acquisition

Patients with advanced solid tumors who failed standard therapy seen in the Early Phase Therapeutics Program clinic were evaluated for tumor tissue and plasma ctDNA by NGS between March 2016 and November 2018. All patients analyzed were Institutional Review Board (IRB) exempt with protocol title “Analysis of Molecular Profiles of Patients with Advanced Cancer” (IRB number: 1804508570), allowing data collection from Caris life sciences and Guardant Health NGS platforms. There were 142 patients paired who had both platform reports. The three patients with Caris reports indicating “tissue with insufficient quantity” were excluded from comparative analysis. Data from platform reports were maintained in a secure network and in secure files. Data from Caris were collected into columns that corresponded to gene alterations of all frequencies, genes with unclassified mutations, specific TP53 alterations, and PD-L1 status. PD-L1 positivity was defined as intensity ≥2+ and ≥5% of immunohistochemically stained cells. Similarly, detected alterations from Guardant360 were collected excluding alterations that were no longer detectable (compared to prior patient plasma samples). A representative sample of patients was highlighted, including all cancer types, proportions, and mutated genes (Table 3).

### 4.2. Cell Fate, Cell Survival, and Genome Maintenance Category Determination

A recent comprehensive review provided categorization of 125 driver genes affected by subtle mutations (Appendix A, Cancer Genome Landscapes) [13]. We integrated this data to define which genes stratify into CF, CS, and GM. Approximately 50 genes detected by Caris and Guardant360, which were not included in this review, were additionally stratified based on descriptions by the National Institute of Health (NIH) Genetics Home Reference database (https://ghr.nlm.nih.gov/). We formulated a guide to designate genes to the appropriate category (bottom of Table 1). Of special note, we classified TP53 as encompassing CS and GM; EP300 and GNAS were each classified in both CS and CF. We also compared the gene category scheme (Appendix A of Cancer Genome Landscapes) to our predictions based on the NIH database. Key attributes of CF included cellular determination; that of CS included promotion of angiogenesis, glucose uptake, and cellular proliferation; and that of GM included DNA repair and stability. Patients’ genes were stratified into these three categories where the value represents the attributable quantity of alterations.

### 4.3. Statistical Analysis

Fisher’s exact test was used for comparison of covariate cohorts to analyze associations and independence. All statistical analyses were done in R. The Fisher.test function in the R stats package was used to assess significance (*p* values). Correction for multiple testing (Q value) was performed using the Benjamini–Hochberg method for the results that had a significant *p*-value. 

### 4.4. Mutation Maps Generation

Mutations in de-identified patients across different cancer subtypes and TP53 alterations detected by both platforms were plotted using Oncoprint. 

### 4.5. Concordance–Discordance Analysis

Only the genes that were shared by both platforms (*n* = 66) were included in the concordance–discordance analysis. Concordance was defined as number of genes that were found to be altered in both platforms. Genes that were found mutated exclusively in Caris or Guardant360 determined discordance. We performed this analysis within each subject and across platforms from the pooled genes of 142 patients. 

## 5. Conclusions

Our comparative analyses of tissue and ctDNA by NGS demonstrated trends in driver and passenger mutations, concordant and discordant genes, and GaDCUS. CS dominated in tumor pathobiology. The utility of treating patients based on the three core cellular processes (CS, GM, and CF) is imperative and requires further evaluation prospectively in clinical trials. In the future, genetic aberration-based cancer genome annotations must extend beyond NGS to proteomic networks [40,41,42]. A comprehensive molecular profile can serve as a guide for the optimal use of off-label drugs, design of relevant clinical trials, and can further the understanding of tumor heterogeneity and evolution to collectively improve patient survival [43]. Preempting tumor evolution via drug-resistance is a major challenge that needs further investigation. Planned serial biopsies of tissue and ctDNA at progression are mandatory in choosing the next best therapy. 

## Figures and Tables

**Figure 1 cancers-12-00701-f001:**
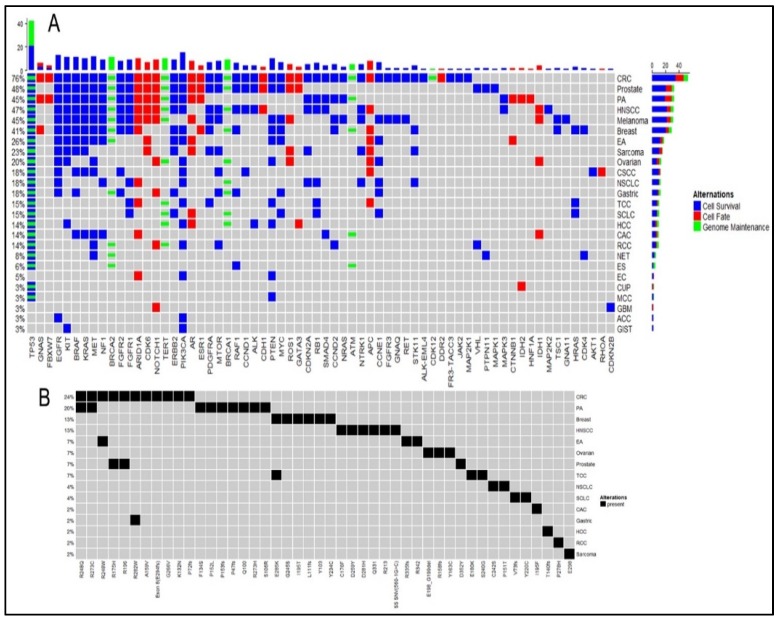
Mutation map (**A**) showing frequency of gene mutations detected by Guardant360 across all and individual cancer subtypes and their associated categories of cell survival, cell fate, and genome maintenance. (**B**) Matched TP53 alterations detected by both Guardant360 and Caris in cancer subtypes.

**Figure 2 cancers-12-00701-f002:**
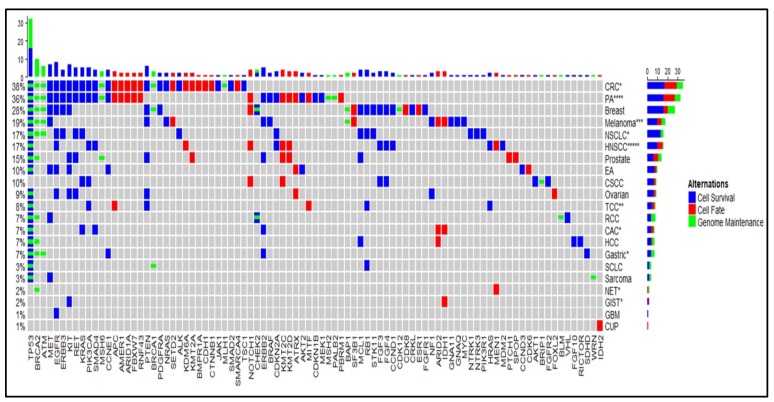
Mutation map showing frequency of gene mutations detected by Caris linked to cancer subtypes and the categories of cell survival, cell fate, and genome maintenance. PD-L1 positive tissue-based IHC found in cancer subtype groups are marked (*) and the number is equivalent to patients.

**Figure 3 cancers-12-00701-f003:**
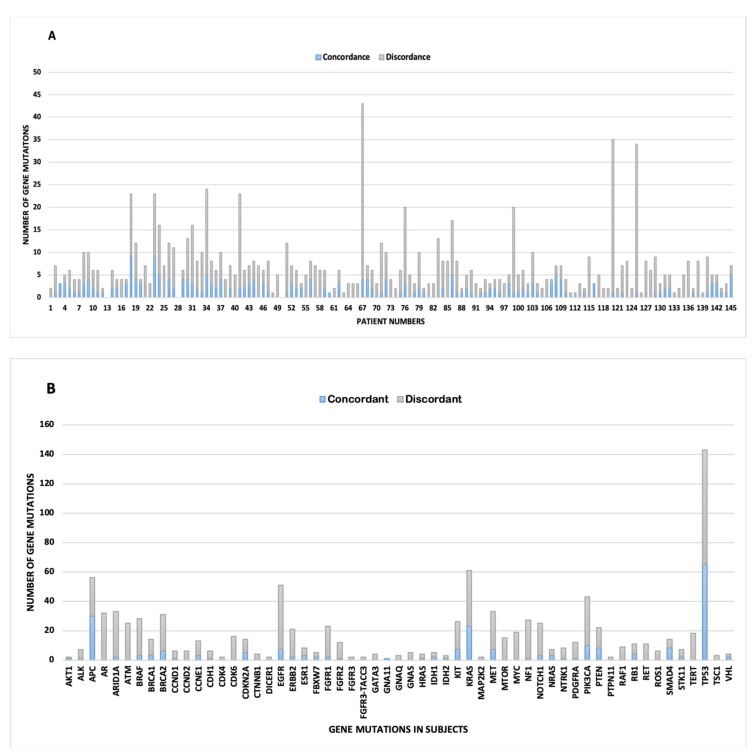
Concordance (blue) and discordance (gray) between gene mutations detected by tissue-based DNA (Caris) and plasma cell-free DNA (Guardant360) next generation sequencing. (**A**) Quantity of gene mutations stack-plotted per de-identified patient number. Discordance shows intra-patient genetic heterogeneity. (**B**) Quantity of gene alterations stack-plotted per mutated genes demonstrates driver and passenger gene mutations that contribute to intra-tumor heterogeneity. Gene mutations that displayed zero concordance and one discordant gene mutation were removed for clarity, and include ARID2, ATRX, DDR2, ERBB3, HNF1A, JAK2, MAP2K1, MLH1, and NTRK3.

**Table 1 cancers-12-00701-t001:** Cancer subtypes and sample size that are stratified in cell fate (CF), cell survival (CS), and genome maintenance (GM) by both next generation sequencing platforms, Caris and Guardant360. Raw values represent quantities of gene mutations per category. Values in parentheses represent gene percentages within the sample group. This table shows major trends that drive tumorigenesis with overall trends at the bottom as the total. Gene designations of CF, CS, and GM also displayed for reference. See appendix for abbreviations.

Cancer Subtype	Q	G-CF (%)	G-CS (%)	G-GM (%)	C-CF (%)	C-CS (%)	C-GM (%)
Adenoid cystic carcinoma	1	0 (0)	2 (100)	0 (0)	0 (0)	0 (0)	0 (0)
Breast adenocarcinoma	10	10 (15.4)	41 (63.1)	14 (21.5)	6 (12.8)	25 (53.2)	16 (34.0)
Carcinoma of unknown primary	2	1 (33.3)	1 (33.3)	1 (33.3)	1 (100)	0 (0)	0 (0)
Cholangiocarcinoma	3	3 (18.8)	9 (56.3)	4 (25.0)	2 (28.6)	4 (57.1)	1 (14.3)
Colorectal carcinoma	29	74 (25.1)	178 (60.3)	43 (14.6)	48 (33.3)	60 (41.7)	36 (25.0)
Cutaneous squamous cell carcinoma	2	3 (18.8)	11 (68.8)	2 (12.5)	4 (36.4)	6 (54.5)	1 (9.1)
Endometrial carcinoma	2	2 (33.3)	4 (66.7)	0 (0)	0 (0)	0 (0)	0 (0)
Esophageal adenocarcinoma	5	5 (14.3)	23 (65.7)	7 (20.0)	2 (12.5)	10 (62.5)	4 (25.0)
Esophageal squamous cell carcinoma	1	0 (0)	2 (33.3)	4 (66.7)	0 (0)	0 (0)	0 (0)
Gastric adenocarcinoma	3	1 (5.9)	13 (76.5)	3 (17.6)	0 (0)	6 (54.5)	5 (45.5)
Gastrointestinal stromal tumor	2	0 (0)	6 (100)	0 (0)	1 (25.0)	3 (75.0)	0 (0)
Glioblastoma multiforme	1	1 (50)	1 (50)	0 (0)	0 (0)	1 (100)	0 (0)
Head and neck squamous cell carcinoma	9	15 (20.3)	48 (64.9)	11 (14.9)	11 (25.6)	22 (51.2)	10 (23.3)
Hepatocellular carcinoma	1	2 (18.2)	5 (45.5)	4 (36.4)	1 (14.3)	4 (57.1)	2 (28.6)
Melanoma	9	10 (16.9)	43 (72.9)	6 (10.2)	6 (25.0)	13 (54.2)	5 (20.8)
Merkel cell carcinoma	1	0 (0)	3 (60.0)	2 (40.0)	0 (0)	0 (0)	0 (0)
Neuroendocrine tumor	2	0 (0)	14 (53.8)	12 (46.2)	1 (50.0)	0 (0)	1 (50.0)
Non-small cell lung cancer	3	2 (9.1)	16 (72.7)	4 (18.2)	0 (0)	20 (83.3)	4 (16.7)
Ovarian carcinoma	6	4 (12.5)	17 (53.1)	11 (34.4)	2 (13.3)	9 (60.0)	4 (26.7)
Pancreatic adenocarcinoma	26	16 (15.0)	70 (65.4)	21 (19.6)	23 (21.5)	58 (54.2)	26 (24.3)
Prostate adenocarcinoma	11	30 (25.6)	68 (58.1)	19 (16.2)	7 (30.4)	8 (34.8)	8 (34.8)
Renal cell carcinoma	3	1 (7.7)	9 (69.2)	3 (23.1)	0 (0)	5 (50.0)	5 (50.0)
Sarcoma	7	4 (12.9)	21 (67.7)	6 (19.4)	0 (0)	3 (60.0)	2 (40.0)
Small cell lung carcinoma	2	1 (6.3)	9 (56.3)	6 (37.5)	0 (0)	4 (57.1)	3 (42.9)
Transitional cell carcinoma	4	2 (8.7)	13 (56.5)	8 (34.8)	2 (13.3)	9 (60.0)	4 (26.7)
TOTAL	145	187 (18.6)	627 (62.4)	191 (19.0)	117 (22.3)	270 (51.5)	137 (26.1)

GENE DESIGNATIONS **Cell fate**: APC, AR, ARID1A, ARID2, ASXL1, ATRX, AXIN1, BCOR, CDH1, CDK6, CDK8, CREBBP, CTNNB1, DAXX, DDR2, DNMT1, DNMT3A, EP300, ESR1, EZH2, FAM123B (AMER1), FBXW7, FOXL2, GATA1, GATA2, GATA3, GNAS, H3F3A, HH, HIST1H3B, HNF1A, IDH1, IDH2, KDM5C, KDM6A, KLF4, KMT2A, KMT2C, KMT2D, MEN1, MITF, MLL3, NF2, NOTCH1, NOTCH2, PAX5, PBRM1, PHF6, PRDM1, PTCH1, RHOA, RNF43, ROS1, RUNX1, SETBP1, SETD2, SF3B1, SMARCA4, SMARCB1, SMO, SPOP, SRSF2, TET2, U2AF1, WT1. **Cell survival:** ABL1, AKT1, AKT2, ALK, ALK-EML4, BCL2, BMPR1A, BRAF, CARD11, CASP8, CBL, CCND1, CCND2, CCND3, CCNE1, CDC73, CDK4, CDKN1B, CDKN2A, CDKN2B, CEBPA, CHEK2, CIC, CRKL, CRLF2, CSF1R, CYLD, DICER1, EGFR, EP300, ERBB2, ERBB3, FGF10, FGF3, FGF4, FGFR1, FGFR2, FGFR3, FGFR3-TACC3, FLT3, FUBP1, GNA11, GNAQ, GNAS, HRAS, JAK1, JAK2, JAK3, KIT, KRAS, MAP2K1, MAP2K2, MAP3K1, MAPK1, MAPK3, MCL1, MDM2, MED12, MEK1, MET, MPL, MTOR, MYC, MYD88, NF1, NFE2L2, NPM1, NRAS, NTRK1, NTRK3, PDGFRA, PIK3CA, PIK3R1, PPP2R1A, PTEN, PTPN11, RAF1, RB1, RET, RICTOR, SDHD, SMAD2, SMAD4, SOCS1, STK11, TGFbR2, TNFAIP3, TP53, TRAF7, TSC1, TSHR, VHL. **Genome maintenance**: ATM, BAP1, BLM, BRCA1, BRCA2, BRIP1, CDK12, CHEK2, MLH1, MSH2, MSH6, PALB2, STAG2, TERT, TP53, WRN.

**Table 2 cancers-12-00701-t002:** Nineteen patients with Guardant360 alterations detected in Caris unclassified section (GaDCUS) and 26 discovered somatic alterations that are potentially treatable. Identified gene mutations show the amino acid alteration in parentheses. Alterations are stratified into the three core cellular process categories to seek trends. Parentheses within the stratified columns represent percentages. See appendix for abbreviations.

PT	Diagnosis	GaDCUS	CF (%)	CS (%)	GM (%)
**5**	Breast adenocarcinoma	HRAS (R41W)	0 (0)	1 (100)	0 (0)
**9**	Breast adenocarcinoma	ARID1A (L1841L)	1 (100)	0 (0)	0 (0)
**23**	Colorectal carcinoma	ALK (F1480L), FGFR3 (A734T), RAF1 (V21M), TERT (A670V)	0 (0)	3 (75)	1 (25)
**26**	Colorectal carcinoma	ARID1A (K1830T)	1 (100)	0 (0)	0 (0)
**33**	Colorectal carcinoma	NF1 (R2119T)	0 (0)	1 (100)	0 (0)
**44**	Colorectal carcinoma	GATA3 (V338I)	1 (100)	0 (0)	0 (0)
**51**	Esophageal adenocarcinoma	AR (P135L)	1 (100)	0 (0)	0 (0)
**56**	Gastric adenocarcinoma	RAF1 (R59H)	0 (0)	1 (100)	0 (0)
**68**	Head and neck squamous cell carcinoma	ALK (G1473E)	0 (0)	1 (100)	0 (0)
**71**	Hepatocellular carcinoma	GATA3 (A319E)	1 (100)	0 (0)	0 (0)
**75**	Melanoma	AR (A810T)	1 (100)	0 (0)	0 (0)
**79**	Melanoma	NTRK1 (G595E), NTRK1 (Q487), ROS1 (G2031R)	1 (50)	1 (50)	0 (0)
**84**	Non-small cell lung carcinoma	RB1 (N690S)	0 (0)	1 (100)	0 (0)
**86**	Non-small cell lung carcinoma	ARID1A (L2239P), ARID1A (R2057W), CCNE1 (A53P)	2 (66.7)	1 (33.3)	0 (0)
**87**	Ovarian carcinoma	ROS1 (P1941L)	1 (100)	0 (0)	0 (0)
**100**	Pancreatic ductal adenocarcinoma	NF1 (R1396H)	0 (0)	1 (100)	0 (0)
**102**	Pancreatic ductal adenocarcinoma	CDKN2A (L64P)	0 (0)	1 (100)	0 (0)
**127**	Prostate adenocarcinoma	MYC (F22L)	0 (0)	1 (100)	0 (0)
**136**	Sarcoma	CDKN2A (A100P)	0 (0)	1 (100)	0 (0)

**Table 3 cancers-12-00701-t003:** Representative data sample of 43 (of 145) patients with accompanying patient numbers and ages. This table displays patients proportional to cancers of both common and rare subtypes. Genetic mutations detected via plasma cfDNA and tissue-based DNA NGS are displayed. Dashes indicate an absence of mutations detected. Time difference indicates gap between sample collections of both platforms in months (DPT). See appendix for abbreviations.

PT	Diagnosis	Age	GDM	CDM	DPT
**1**	Adenoid cystic carcinoma	87	EGFR, PIK3CA	-	3
**3**	Breast adenocarcinoma	36	EGFR, PTEN, TP53	EGFR, PTEN, TP53	11
**4**	Breast adenocarcinoma	51	PTEN, FGFR2, FGFR1, KRAS, PIK3CA, TP53, BRCA2	BRCA2, CDK8, PTEN, TP53	1
**11**	Breast adenocarcinoma	71	ARID1A, GNAS, TP53	ATM, BAP1, BRCA2, NOTCH1, TP53	1
**12**	Carcinoma of unknown primary	27	IDH2, TP53	IDH2	1
**15**	Cholangiocarcinoma	83	ATM, BRAF, SMAD4, TP53	SMAD4, TP53	8
**17**	Colorectal carcinoma	31	APC (x2), TP53, ARID1A	APC (x2), TP53	3
**21**	Colorectal carcinoma	76	NF1, ROS1, STK11	APC, KIT, KRAS, TP53	1
**22**	Colorectal carcinoma	58	APC, KRAS, TP53	-	0
**35**	Colorectal carcinoma	63	APC (x2), AR, EGFR, FGFR1, KRAS, PIK3CA	APC, ATM, KRAS, PIK3CA	13
**42**	Colorectal carcinoma	56	APC, KRAS, MET, MYC, RAF1, TP53	APC, TP53	0
**46**	Cutaneous squamous cell carcinoma	72	CCND1, EGFR, FGFR2, KRAS, PIK3CA	BRIP1, KRAS, NOTCH1, PIK3CA; FGF3, FGF4, FGFR2, NOTCH1	2
**48**	Endometrial carcinoma	71	PIK3CA	-	57
**54**	Esophageal adenocarcinoma	66	KIT, TP53	AKT2, KIT, CDK6, TP53	38
**57**	Gastric Adenocarcinoma	52	CCNE1 (x2), RAF1	ATM, BRCA2, Her2/Neu (ERBB2), TP53	5
**60**	Gastrointestinal stromal tumor	70	KIT	KIT	0
**61**	Glioblastoma	63	CDKN2B, NOTCH1	EGFR	9
**63**	Head and neck squamous cell carcinoma	63	TP53	KMT2D	25
**69**	Head and neck squamous cell carcinoma	63	CCND1, CDH1, MET, PDGFRA, PIK3CA, TP53	CCND1, FGF3, FGF4, KMT2C (x2), TP53	5
**71**	Hepatocellular carcinoma	63	ALK, AR, BRCA1, GATA3, KIT, PIK3CA, PTEN, TERT (x2), TP53	ARID2, FGF10, TP53, BRCA2, MCL1, RICTOR	3
**72**	Melanoma	36	BRAF (x2), EGFR, MET (x2), NF1, NRAS, PTEN, TERT (x2)	BRAF, MET, NF1, PTEN	2
**78**	Melanoma	68	GNA11, MYC, NOTCH1	BAP1, GNA11	6
**80**	Melanoma	70	ARID1A, NRAS	NRAS, SF3B1	4
**81**	Merkel cell carcinoma	80	PTEN, TP53 (x2)	-	0
**82**	Neuroendocrine tumor	68	MET, PTPN11	BRCA2	50
**84**	Non-small cell lung carcinoma	71	CCNE1, CDKN2A, EGFR, FGFR1, NF1, PIK3CA, RB1, TP53	EGFR, PIK3R1, TP53	1
**88**	Ovarian carcinoma	57	BRCA1, TP53	TP53	0
**90**	Ovarian carcinoma	77	CCNE1, EGFR, NOTCH1, PIK3CA, PTEN	PTEN, TP53	1
**93**	Pancreatic ductal adenocarcinoma	53	KRAS, NF1, TP53	CDKN2A, KRAS	8
**94**	Pancreatic ductal adenocarcinoma	72	CTNNB1, KRAS, TP53	KRAS, TP53	11
**95**	Pancreatic ductal adenocarcinoma	61	ARID1A, CDKN2A, KRAS, TP53	KRAS, RNF43, TP53	9
**110**	Pancreatic ductal adenocarcinoma	67	CDK6, FBXW7, KRAS	AMER1, KRAS, PALB2, SMAD4	1
**116**	Pancreatic ductal adenocarcinoma	55	CDKN2A, KRAS, TP53	CDKN1B, CDKN2A, KRAS, TP53	4
**122**	Prostate adenocarcinoma	51	AR, BRAF, CDK6, FGFR1, MET, RAF1, TP53	TP53	17
**126**	Prostate adenocarcinoma	56	TP53	-	1
**129**	Prostate adenocarcinoma	67	AR, BRAF, BRCA2, CDK6, MET, MYC, PDGFRA, ROS1, TP53	TP53	0
**131**	Renal cell carcinoma	52	BRCA2 (x2), CCND2, TERT, VHL	BRCA2, VHL	0
**133**	Sarcoma	38	-	MET	7
**137**	Sarcoma	20	MTOR, TP53	TP53	6
**139**	Sarcoma	41	TP53	WRN	0
**141**	Small cell lung carcinoma (SCLC)	60	BRCA1, HRAS, RB1, TP53 (x2)	BRCA1, RB1, TP53	2
**143**	Transitional cell carcinoma	63	PIK3CA, TP53	MITF, PIK3CA	20
**144**	Transitional cell carcinoma	83	FGFR1, TERT, TP53	TP53	2

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
