# Peer review of "A Comparative Analysis of Tumors and Plasma Circulating Tumor DNA in 145 Advanced Cancer Patients Annotated by 3 Core Cellular Processes"

_cancers, 2020, doi:10.3390/cancers12030701_

Round 1
Reviewer 1 Report
Data and Analysis Summary:
The data presented in "A Comparative Analysis of Tumor and Plasma Circulating Tumor DNA in 145 Advanced Cancer Patients Annotated by 3 Core Cellular Processes," represents a significant effort in the collection of translational genomic data collection, including a relatively large albeit heterogeneous population of patients with cancers characterized by commercial assays for NGS sequencing of tumor biopsy sample as well as ctDNA. Furthermore, the authors stratify the diverse alterations identified into 3 classes related to cellular functionality of the altered gene in question. Graphical network analysis is applied to display the results of the two techniques as related to the 3 mutational classes.
Summary of Results and Conclusions:
The authors identify discordant mutation calls for individual patients using the two techniques. The authors identify 26 mutations identified in by both techniques which they consider to be somatic, and possible actionable.
Barriers to Publication:
The data are presented with essentially no statistical analysis or statistical comparison between groups. The 3 core cellular processes seem to be based on the literature, but other than CS mutations being most common, I do not see how this is useful. Does it separate patient, tumor or technical groups? If so this should be demonstrated statistically. Much attention is given to "interactomes," but their utility again seems poorly justified by the report. The connectivity represented by the presented networks is uncertain. More specifically, what is an edge (ie. line) in these networks? What is the advantage of a network analysis approach in this case? Networks are the subject of a rich set of mathematical descriptors which can be leveraged to describe, compare and subdivide networks - If network theory is useful to describe these data, at lest some numerical analysis of the presented graphs/networks should be presented. Such an analysis may indeed help with the statistical comparison of groups which is lacking in this draft. The discordance (rates) between techniques is interesting and a useful and unique component of this dataset. However, again no statistical analysis is provided to justify if the difference is real / detectible. This could likely be achieved by making some reasonable groupings of patients / tumor types. Were some gene more disconcordant than others to an extent that it can be detected statistically? The discussion of the 26 "actionable" mutations needs to be improved. Please describe how one is sure these mutations are somatic based on scientific first principles - can just be from the companies reference materials, but logical justification needs to be provided. Also what is meant by actionable - are there targeted therapies for these?Summary:
Unfortunately I can not recommend publication at this time, but I think you do have an interesting and useful data set. I congratulate you on putting it together. However, a statistical rather than descriptive approach is needed; and attempting to make comparison between subsets of the data (even if negative results) will be key to making this publishable. Graph/network analysis of data should be avoided engaged quantitatively.
Author Response
Thank you for taking the time to review our study. Please see our responses below which correspond to the updated version of our manuscript. We are grateful for your feedback to improve our study. Please do not hesitate to ask us any questions.
Point 1: The data are presented with essentially no statistical analysis or statistical comparison between groups. The 3 core cellular processes seem to be based on the literature, but other than CS mutations being most common, I do not see how this is useful. Does it separate patient, tumor or technical groups? If so this should be demonstrated statistically. The discordance (rates) between techniques is interesting and a useful and unique component of this dataset. However, again no statistical analysis is provided to justify if the difference is real / detectible. This could likely be achieved by making some reasonable groupings of patients / tumor types. Were some gene more discordant than others to an extent that it can be detected statistically? The discussion of the 26 "actionable" mutations needs to be improved. Please describe how one is sure these mutations are somatic based on scientific first principles - can just be from the companies’ reference materials, but logical justification needs to be provided. Also what is meant by actionable - are there targeted therapies for these?
Response 1: As recommended by the reviewer, we have conducted further analyses of the cohort by dividing them into different groups and testing for significance. This includes 1) age (>60yr vs <60yr) association in cohorts with respect to the three core cellular processes, 2) TP53 mutational status (+ve vs –ve) in patient cohorts and its influence on mutational patterns for the three core processes, 3) age specific cohorts and any association with TP53 mutational status, 4) co-dependence of the three core processes (CS, CF and GM in pairwise manner).
Fisher exact test was done by building 2X2 contingency tables for each of these comparisons. We looked at the comparisons across all tumor types. The results are included in the text and the tables are provided as Supplemental tables 1A, B and C.
The Caris-MI/X NGS platform analyzes only tumor exon mutations in oncogenes and tumor suppressors. In contrast, the Gaurdant360 NGS platform analyzes cfDNA tumor vs. normal donor volunteer WES (age 20-40 years) i.e. reference normal DNA, mentioned on page 10 lines 258 to 262. Plasma cfDNA from patients with mutations can detect up to 0.1% mutant allele frequencies (MAFs) from a background of cfDNA extracted from healthy donors are reported as acquired somatic mutations [Lanman RB et al. PLoS One, 2015] by digital sequencing algorithms. Guardant Health only reports germline mutations for BRCA 1 and 2 genes [34].
An actionable mutation is defined as a genetic aberration in the DNA (e.g. activating mutation) when detected in a patient's tumor, would be expected or predicted to affect a response to a targeted treatment [Carr TH et al. Nat Rev Cancer, 2016] available in basket or umbrella clinical trials or FDA-approved or be available for off-label treatment. We have included this information in the text page 8 line 180.
Point 2: Much attention is given to "interactomes," but their utility again seems poorly justified by the report. The connectivity represented by the presented networks is uncertain. More specifically, what is an edge (ie. line) in these networks? What is the advantage of a network analysis approach in this case? Networks are the subject of a rich set of mathematical descriptors which can be leveraged to describe, compare and subdivide networks - If network theory is useful to describe these data, at lest some numerical analysis of the presented graphs/networks should be presented. Such an analysis may indeed help with the statistical comparison of groups which is lacking in this draft.
Response 2: We agree with the reviewer. The network figures were representing associations of gene mutations across tumor types and other annotations. We now are providing the same information as mutational maps with all the associations included in the networks presented with much better visualization that includes frequency of mutation across tissue types and their representative association with the three cellular processes. Both the figures 1 and 2 have been modified.
Reviewer 2 Report
This is a interesting paper with some relevant results. However some points need to be addressed
1) Please provide pvalues when comparing both platfforms, to be sure there are differences among them
2) table 2 is a bit tricky. How were the patients selected?? Which was the first sample collected? please add this info
3)Authors could eliminate from the study, samples collected within difference like for example of more tan one year and try to make the analysis again, to test if there are still differences.
